# Delamination Behavior of CFRP Laminated Plates under the Combination of Tensile Preloading and Impact Loading

**DOI:** 10.3390/ma16196595

**Published:** 2023-10-08

**Authors:** Kaiwei Lan, Haodong Wang, Cunxian Wang

**Affiliations:** 1School of Aeronautics, Northwestern Polytechnical University, Xi’an 710072, China; lankaiwei2023@163.com (K.L.); wanghaodong724@mail.nwpu.edu.cn (H.W.); 2Institute of Extreme Mechanics, Northwestern Polytechnical University, Xi’an 710072, China; 3Shaanxi Key Laboratory of Impact Dynamics and Its Engineering Application (IDEA), Northwestern Polytechnical University, Xi’an 710072, China

**Keywords:** biaxial tensile preload, delamination, CFRP laminated plate, competition mechanism, impact loading

## Abstract

When subjected to impact loading, aircraft composite structures are usually in a specific preloading condition (such as tension and compression). In this study, ballistic tests were conducted using a high-speed gas gun system to investigate the effect of biaxial in-plane tensile preload on the delamination of CFRP laminates during high-speed impact. These tests covered central and near-edge locations for both unloaded and preloaded targets, with the test speeds including 50 m/s, 70 m/s, and 90 m/s. The delamination areas, when impacting the center location under 1000 με, show a 14.2~36.7% decrease. However, the cases when impacting the near-edge location show no more than a 19.3% decrease, and even more delamination areas were observed. In addition, in order to enhance the understanding of experimental phenomena, numerical simulations were conducted using the ABAQUS/Explicit solver, combined with the user subroutine VUMAT with modified Hou criteria. The experimental and simulation results were in good agreement, and the maximum error was approximately 12.9%. The results showed that not only the preloading value but also the impact velocity have significant influences on the delamination behavior of preloaded CFRP laminated plates. Combining detailed discussions, the biaxial tensile preload enhanced the resistance to out-of-plane displacement and caused laminate interface stiffness degradation. By analyzing the influence of the preloading value and impact velocity on competing mechanisms between the stress-stiffening effect and interface stiffness degradation effect, the complex delamination behaviors of laminates under various preloading degrees and impact velocities at different impact locations were reasonably explained.

## 1. Introduction

Carbon-fiber-reinforced polymer (CFRP) composites present many excellent mechanical properties compared to traditional metal materials, making them widely used in aircraft [1]. In the daily usage of civil aircraft, composites may produce damage when suffering various forms of impact loading, such as the impact of birds, gravel, falling tools, and hail. The predominant impact-induced damages primarily encompass fiber fracture and matrix crack, with the latter often leading to non-apparent delamination, invisible to the unaided eye. However, delamination significantly affects the integral stiffness and strength of composite structures, which may finally result in serious security threats to aircraft.

Over the years, investigations on the impact response of composites have attracted extensive attention from many scholars, and a number of good papers have been published. Due to the limited science and technology available in its early days, the original research mainly focused on ballistic tests and damage measurements. Lee et al. [2] and Cheng et al. [3] carried out a series of impact tests on composites, and the results showed that the typical damage of composites mainly comprised crushing, fiber tensile fracture, and delamination. In addition, the ballistic limits of laminates were determined, and the effect of the projectile shape on perforation was also discussed. Yew and Kendrick [4] introduced ultrasonic C-scan and cross-section staining methods to detect internal damage in laminates under impact loads, revealing the anisotropic nature of composite impact damage. With the development of finite element (FE) technology, more and more scholars have attempted to investigate this topic by combining tests and simulations. Binienda and Zhang et al. [5,6] devised an array of experimental approaches and multi-scale numerical methodologies. These were employed to comprehensively examine the mechanical responses of triaxial braided composites, effectively establishing the impact threshold and delineating the various mechanisms governing damage patterns. Camanho et al. [7], as well as Sridharan and Pankow [8], posited that cohesive elements exhibit the capability to forecast the initiation and extension of delamination in composite materials. Rajaneesh et al. [9] formulated finite element models to explore the high-velocity impact responses of composite materials, introducing a physically grounded delayed damage model in their research. Furthermore, Xu et al. [10] conducted ballistic impact tests on CFRP plates under three different conditions: preloaded uniaxial tension, preloaded uniaxial compression, and no preload. They analyzed the energy absorption efficiency and failure modes and established a theoretical model based on energy absorption theory. Meanwhile, Qaderi, Ebrahimi, and Vingas [11] conducted a dynamic theoretical analysis of multi-layer composite beams reinforced with GPLs using the high-order shear deformation beam theory. They obtained the trends in the natural frequencies of the system as a function of the system parameters for different distribution patterns.

Nowadays, the majority of existing studies primarily concentrate on CFRP laminated plates without prior preloading. Notably, the reviews by Abrate [12] and the investigations by Reid and Zhou [13] predominantly emphasize the impact behavior of unloaded composite structures. Consequently, there exists a limited understanding of the impact properties of composites subjected to preloading conditions. However, composite structures on aircraft are usually in a specific loading condition (such as tension or compression) when subjected to impact loading [1]. For instance, during aircraft takeoff and landing maneuvers, the wing may undergo upward bending, potentially resulting in impact from debris on the runway. During this phase, the lower skin structure is subjected to significant tensile loads. Similarly, the upper skin of the wing can encounter impacts from hailstones, leading to substantial compressive loads on the upper skin structure. Additionally, engine blades may experience significant centrifugal dynamic loads and aerodynamic forces when impacted by flying birds. Hence, investigating the influence of preloads on the impact characteristics of composite laminates becomes imperative.

As an illustration, fuselage skins in aircraft service frequently encounter operational strains reaching up to 1500 με [11]. NASA held a seminar and put forward that the preloading effect was critical to the impact damage of composites. The LIBCOS and MAAXIMUS projects launched by EASA and Advisory Circular 20-107B of the FAA also stated that the actual tensile or compressive preloads should be considered when analyzing accidental impact damage. Williams et al. [14] pointed out that the preload’s effect on composite structures’ impact behavior is a potential problem in aircraft certification. However, most research focuses on the impact behavior of unloaded composites, and it is vital to study the impact responses of composites under specific load conditions, especially under preloading conditions.

At present, some studies have been devoted to the experimental and numerical evaluation of the impact behavior of preloaded composites. Both low-velocity and high-velocity impact tests on preloaded composites have been carried out by using drop-weight test instruments [14,15,16], impact pendulums [17], and gas gun devices [18,19]. Moreover, Mikkor et al. [20] and Pickett et al. [21] conducted explicit FE simulations on the impact process of unloaded and preloaded laminates using Pam-Crash. Heimbs et al. [1,22] simulated preloading and impact using a “layered shell” model and “implicit explicit coupling” in LS-DYNA and ABAQUS, respectively. It should be pointed out that there existed significant differences between these presented research studies. Garnier [19], Chiu [23], and Kelkar [24] proposed that tensile preload leads to an increase in the impact damage area for laminates. However, the works by Heimbs [22], Robb [25], García Castillo [26], Zhikharev [27], and Guillaud et al. [28] showed that tensile preload can reduce the delamination area. In addition, investigations by Mitrevski [29], Moallemzadeh [30], and Choi et al. [31] indicated that tensile preloading levels do not have an effect on impact damage under low-velocity impact loading. Wang et al. [32] found different delamination tendencies when impacting two different locations and that the interlaminar stresses induced a weakening effect on delamination. While these investigations present reasonable findings, further experimental, numerical, and theoretical research is needed to understand how preloads influence the impact response of composite laminates under different preloading levels and impact velocities.

In the present work, a series of ballistic impact tests were carried out on CFRP laminated plates with two different stacking sequences to understand this topic further. Three preloading levels (0 με, 500 με, and 1000 με) were realized using a specially designed preloading device. Moreover, three different impact velocities (50 m/s, 70 m/s, 90 m/s) and two typical impact locations (center location and near-edge location) were considered in the ballistic impact tests. Additional numerical simulations were also conducted in ABAQUS/Explicit to gain further insights into the effects of preloading levels and impact velocities on the delamination behavior of CFRP laminated plates under impact loading. Based on the comparison of experimental and numerical results, the influence of the preloading degree and impact velocity on the competing mechanisms between the stress-stiffening effect and interface stiffness degradation effect was analyzed.

## 2. Material Preparation and Experimental Procedure

### 2.1. Material Preparation, Preloading Fixture, and Test Conditions

The CFRP composite specimens tested in the ballistic impact tests were made of T700/epoxy resin M10R laminated plates. The laminated plates were prepared to consist of 16 plies with two types of stacking sequences ([0/90]_8_ and [0/90/+45/−45]_2S_), and each panel was molded in one shot to produce a nominal fiber volume fraction of 58%. All specimens were prepared with a size of (300 × 300) mm^2^ by using water jet cutting to avoid initial damage as much as possible during processing. Figure 1 shows the preloading fixture assembly, which consists of eight steel clamping pieces, four thick steel frames, and an extensive steel support. Each side of the specimen was clamped by two clamping pieces, thus leaving a 210 × 210 mm^2^ free impact zone. A set of eight strain gauges was affixed to the rear surface of each panel, serving a dual purpose: recording in-plane strain histories during impact and providing pre-strain control feedback prior to impact. With the support of these strain gauges, the specific pre-strain could be accurately applied by adjusting the bolts connecting the clamping pieces with the frames before the impact tests. In the present work, a series of ballistic impact tests were carried out under different preloads (0 με, 500 με, and 1000 με) and impact velocities (50 m/s, 70 m/s, and 90 m/s). In addition, two impact positions (point 1 and point 2) were predetermined for the ballistic tests in which point 1 was located in the center of the target and point 2 was 70 mm apart from point 1 in the horizontal direction, as shown in Figure 1. For each loading condition, at least three valid tests were performed. More than 120 pieces of CFRP laminated plates were consumed in the ballistic tests.

### 2.2. Ballistic Impact Test Devices

Figure 2 illustrates the one-stage compressed gas gun system used for the high-speed impact tests for preloaded CFRP laminated plates. The gas gun system comprised a pressure vessel with a volume of 0.2 m³ and a 6.8 m long gun barrel with an inner diameter of 80 mm. As projectiles, steel spheres with a diameter of 10 mm and a corresponding mass of 4.05 g were chosen. Specially designed 3D-printed PLA shells were used to support the projectiles and were accelerated by compressive air in the gun barrel during the impact test, contributing to the control of the attitude and velocity of projectiles. Two high-speed cameras were positioned in front of the CFRP laminated plate for specific purposes: one camera was dedicated to recording the impact position, while the other was utilized to measure velocity. In addition to using high-speed cameras, the impact velocities of the projectiles were also measured using a laser velocimeter, which helped us to obtain reliable and accurate impact velocities. After the impact tests, the specimen’s visible failures were evaluated first, and then the internal delamination damages were detected using a PAC ultrasonic C-scan.

## 3. Experimental Results and Analysis

Figure 3a,b show the typical front-view high-speed photographs of point 1 and point 2 on the CFRP laminated plates impacted by steel sphere projectile under the speed of 50 m/s, respectively. Via the statistical analysis of all experimental data, the deviations between the actual impact positions and predetermined positions do not exceed 9.8 mm, and the deviations in the impact speeds do not exceed 4 m/s. The CFRP laminated plates present smooth circular dents in different degrees on the front side under impact velocities of 70 m/s and 90 m/s, while there is no visible dent in the case of 50 m/s. Furthermore, there was no observable damage detected on the rear side of any of the targets subjected to impact velocities of 50 m/s, 70 m/s, and 90 m/s. The delamination areas of the CFRP laminated plates subjected to various biaxial tensile preloads and impact speeds are summarized in Table 1 and Figure 4.

As vividly shown in Table 1 and Figure 4, for the CFRP laminated plates subjected to the same impact loading, the delamination areas show a decreasing tendency as the biaxial in-plane tensile pre-strains increase when impacting the center location (point 1). Moreover, the typical in-plane strain histories during impact for the CFRP laminated plates under various biaxial tensile preloads are illustrated in Figure 5a–c, and the strain amplitudes under various biaxial tensile preloads at different impact velocities (50 m/s, 70 m/s, and 90 m/s) are summarized in Figure 5d–f, based on the in-plane strain histories. The strain amplitudes at all three impact velocities show a decreasing tendency with the increasing biaxial in-plane tensile pre-strains.

Theoretically, when the projectile impacts the CFRP laminated plate, two kinds of stress waves (transverse wave and longitudinal wave) are generated at the impact point. These then propagate in the laminated plate along the in-plane direction. The longitudinal wave only induces longitudinal strain without changing the shape of the CFRP laminated plate. In contrast, the transverse wave only causes the shape of the CFRP laminated plate to change but does not produce longitudinal strain. Since the longitudinal wave propagates outward with a higher speed than the transverse wave [33], the target plate component passing through the transverse wavefront usually presents with a “V” shape, as shown in Figure 6a,b. Figure 6c,d illustrates the enhancement mechanism of the resistance to out-of-plane displacement for CFRP laminated plates when applying in-plane tensile preloads. As shown, the unloaded CFRP laminated plate produces resistance force ($2\cdot {F^{\prime }}_{t}\cdot \sin\theta^{\prime }$) when suffering impact loading, and such resistance force is derived from the fiber tensile load. In comparison, the laminated plate provides additional resistance force ($2\cdot F_{p}\sin\theta$) when applying in-plane tensile preloads. It requires more significant impact loading to achieve the same bending degree for the preloaded CFRP laminated plate, proving that applying in-plane tensile preloads enhances the resistance to out-of-plane displacement. That is, when suffering the same impact loading, there will exist a reduction in the bending degree for preloaded targets in contrast to unloaded targets, as shown in Figure 6c,d. Correspondingly, the delamination propagation of CFRP laminated plates under impact loading is prevented indirectly when applying in-plane tensile preloads. Therefore, the biaxial in-plane tensile preloads are supposed to play a positive role in delamination resistance when impacting the center location of CFRP laminated plates, and such a positive effect is defined as the stress-stiffening effect.

In addition, the delamination areas impacting the near-edge location (point 2) of preloaded CFRP laminated plates are significantly higher than those impacting the center location (point 1). For both [0/90]_8_ and [0/90/+45/−45]_2S_ laminated plates under 50 m/s impact loading, the preloaded targets even show more delamination than unloaded targets, which can also be observed in the previous work [32]. However, the strain amplitudes (seen in Figure 5d–f) still decrease as the biaxial in-plane tensile pre-strains increase, which is almost the same as when impacting the center location of the targets. Moreover, the energy absorption rate is introduced to further prove the existence of the stress-stiffening effect when impacting the near-edge location (point 2). During the impact process, most of the initial kinetic energy of the projectile is absorbed by the target plate in the form of local damage and global deformation, and a small part of the energy is transferred to the rebound kinetic energy of the projectile. Therefore, the energy of the projectile absorbed by the target can be assumed as
(1)$$E=E_{0}-E_{r}=\frac{1}{2}m\cdot (v_{0}{}^{2}-v_{r}{}^{2})$$
where $E_{0}$ and $E_{r}$ denote the initial kinetic energy and rebound kinetic energy of the projectile, and m is the mass of the projectile. $v_{0}$ is the initial impact velocity of the projectile which can be obtained using the laser velocimeter. However, it is difficult to measure the rebound velocity of the projectile $v_{r}$. Figure 7 shows typical photographs of the impact process and rebound process of the projectiles taken using the high-speed camera. As shown, the relations between $v_{0}$ and $v_{r}$ can be depicted as
(2)$$\frac{v_{0}}{v_{r}}=\frac{N_{1}+N_{2}+N_{3}+N_{4}}{n_{1}+n_{2}+n_{3}+n_{4}}$$
where $N_{i}(i=1,2,3,4)$ and $n_{i}(i=1,2,3,4)$ represent the pixel numbers that are used for depicting the distances between the projectile positions at even intervals. Based on the measurements obtained from the high-speed camera and laser velocimeter, the initial impact velocity and rebound velocity of the projectile can be accurately measured, thus contributing to the reliable calculation of the energy of the projectile absorbed by the target. Thus, the energy of the projectile absorbed by the target E can be written as
(3)$$E=\frac{1}{2}mv_{0}{}^{2}\cdot \left[1-{\left(\frac{n_{1}+n_{2}+n_{3}+n_{4}}{N_{1}+N_{2}+N_{3}+N_{4}}\right)}^{2}\right]$$

The energy absorption rate α can be defined as
(4)$$\alpha =\frac{E}{E_{0}}=1-{\left(\frac{n_{1}+n_{2}+n_{3}+n_{4}}{N_{1}+N_{2}+N_{3}+N_{4}}\right)}^{2}$$

Figure 8 shows the energy absorption rates when impacting the center location (point 1) and near-edge location (point 2) of unloaded/preloaded CFRP laminated plates at impact velocities of 50 m/s, 70 m/s, and 90 m/s. As shown, for both [0/90]_8_ and [0/90/+45/−45]_2S_ laminated plates, the energy absorption rates still present decreasing tendencies as the biaxial in-plane tensile pre-strains increase when impacting the near-edge location, which is similar to the cases in which the center location is impacted. Therefore, the resistance to out-of-plane displacement for CFRP laminated plate is still enhanced by biaxial in-plane tensile preloads, and such stress-stiffening effect still has a positive effect on the delamination resistance when impacting the near-edge location (point 2) of CFRP laminated plates.

The interface stiffness degradation effect is supposed to have a negative effect on the delamination resistance of preloaded CFRP laminated plates. Generally, the laminated plate usually presents tensile-shear coupling or the mismatch of Poisson’s ratio due to the different ply orientations, thus resulting in the generation of interlaminar normal stress and interlaminar shear stresses between each two adjacent layers when applying in-plane tensile preloads [31]. Such interlaminar stresses cause the degradation of the interface stiffness of CFRP laminated plates to different degrees. Therefore, the interfacial strengths of the laminated plates are weakened by the preloads induced by the interface stiffness degradation, which makes delamination even easier to generate and propagate in the laminated plates when subjected to impact loading. The previous work [32] proposed that competing mechanisms exist between the stress-stiffening effect and the interface stiffness degradation effect and influence the delamination behavior of preloaded CFRP laminated plates. Both the stress-stiffening effect and the interface stiffness degradation effect are due directly to the application of biaxial in-plane tensile preloads. Moreover, interface stiffness degradation is to a higher degree at the near-edge location than that at the center location, which could explain why the delamination areas when impacting the near-edge location (point 2), are significantly higher than those when impacting the center location (point 1).

It deserves to be noticed that the experimental results indicate that the preloading value and impact velocity have a significant influence on the competition between the stress-stiffening effect and the interface stiffness degradation effect. On the one hand, the decreased delamination areas also tend to increase as the preloading values increase when impacting the center location. On the other hand, the delamination areas of target plates under 1000 με biaxial tensile preloads are always fewer than those under 500 με biaxial tensile preloads, whether impacting the center location or the near-edge location. In the following sections, numerical modeling is carried out to gain further insights into the effect of the preloading value and impact velocity on the competition between the stress-stiffening effect and the interface stiffness degradation effect.

## 4. Numerical Modeling and Validation

### 4.1. Finite Element Modeling

The high-velocity impact tests conducted on both unloaded and preloaded CFRP laminated plates were simulated using the ABAQUS/Explicit solver. Figure 9 shows the mesh for the L × W = (300 × 300) mm^2^ panel configuration with projectile. Given the widespread adoption of the 8-node hexahedral cohesive elements introduced by Hillerborg et al. [21] for predicting delamination in composite materials, a finite element model was constructed. This model comprised 16 three-dimensional solid plies, each with a thickness of 0.125 mm, and 15 cohesive plies, each with a thickness of 0.001 mm. It was employed to forecast the impact-induced damage in both unloaded and preloaded laminated plates. In detail, 242,064 three-dimensional elements with reduced integration were used to model the composite material, and 226,935 cohesive elements were applied to model the interface. An investigation into size dependency was conducted to identify an optimized mesh for impact analysis. This involved determining the most suitable mesh size through a comparison of numerical results and experimental data. The failure loads, overall deformation, and crack tip position, as predicted through the modification of the interfacial strength, exhibited a high degree of independence from mesh-related variations when the element sizes were maintained below 3 mm in Turon’s work [34], so the element size was finally set as 2 mm. In addition, the contact behaviors between the projectile and the target were set using “General contact” combined with “surface contact” pairs. To perform an accurate contact analysis between the projectile and laminated plate, the projectile was set as a rigid body, and its mesh was refined to match the element size of the laminated plate. Additionally, in the simulations, a viscosity-based stabilization method was implemented to mitigate the occurrence of hourglass modes within the reduced integration elements.

### 4.2. Material Modeling

To anticipate the deformation and damage of CFRP laminated plates, whether unloaded or preloaded, when subjected to a combination of biaxial in-plane tensile preloads and impact loads, we employed a user subroutine VUMAT. This subroutine relies on an orthotropic material model and incorporates the revised Hou criterion [35]. This revised Hou criterion is a strain-based criterion and has been proven to satisfy the modeling of composite laminated plates under impact loading in previous work [36]. In the present work, the failure criterion is minorly modified as follows:(1)Fiber failure: (5)$$e_{f}={\left(\frac{\varepsilon_{11}}{\varepsilon_{X_{T}}}\right)}^{2}+{\left(\frac{\varepsilon_{12}}{\varepsilon_{S_{f}}^{12}}\right)}^{2}+{\left(\frac{\varepsilon_{13}}{\varepsilon_{S_{f}}^{13}}\right)}^{2}\geq 1$$(2)Matrix cracking ($\sigma_{22}\geq 0$):(6)$$e_{mt}^{}={\left(\frac{\varepsilon_{22}}{\varepsilon_{Y_{t}}}\right)}^{2}+{\left(\frac{\varepsilon_{12}}{\varepsilon_{S_{12}}}\right)}^{2}+{\left(\frac{\varepsilon_{23}}{\varepsilon_{S_{m23}}}\right)}^{2}\geq 1$$(3)Matrix crushing ($\sigma_{22}+\sigma_{33}<0$):(7)$$e_{mc}^{}=\frac{1}{4}{\left(\frac{\varepsilon_{22}}{\varepsilon_{S_{12}}}\right)}^{2}+\frac{Y_{c}{}^{2}\varepsilon_{22}}{4\varepsilon_{S_{12}}{}^{2}\varepsilon_{Y_{c}}}-\frac{\varepsilon_{22}}{\varepsilon_{Y_{c}}}+{\left(\frac{\varepsilon_{12}}{\varepsilon_{S_{12}}}\right)}^{2}\geq 1$$
where
(8)$$\varepsilon_{X_{t}}=\frac{X_{t}}{E_{11}},\varepsilon_{Y_{t}}=\frac{Y_{t}}{E_{22}},\varepsilon_{Y_{c}}=\frac{Y_{c}}{E_{22}},\varepsilon_{S_{12}}=\frac{S_{12}}{G_{12}},\varepsilon_{S_{m23}}=\frac{S_{m23}}{G_{23}},\varepsilon_{S_{f}}^{12}=\frac{S_{f}}{G_{12}},\;\varepsilon_{S_{f}}^{13}=\frac{S_{f}}{G_{13}}$$


$e_{f}$, $e_{mt}$, and $e_{mc}$ denote the parameters for specific damage; $\varepsilon_{11}$, $\varepsilon_{22}$, and $\varepsilon_{33}$ are the instant strains in the fiber, transverse, and through-thickness direction; and $\varepsilon_{12}$, $\varepsilon_{13}$, and $\varepsilon_{23}$ are the instant shear strains. $\varepsilon_{X_{t}}$, $\varepsilon_{Y_{t}}$, $\varepsilon_{Y_{c}}$, $\varepsilon_{S_{12}}$, $\varepsilon_{S_{m23}}$, $\varepsilon_{S_{f}}^{12}$, and $\varepsilon_{S_{f}}^{13}$ represent the strain limits and shear strain limits. $X_{t}$ and $Y_{t}$ denote the tensile strength in the fiber and transverse direction, $Y_{c}$ is the compressive strength in the fiber and transverse direction, $S_{12}$ is the shear strength in the plane of the fiber and the transverse direction, $S_{m23}$ is the shear strength for matrix cracking in the plane of transverse and through-thickness direction, and $S_{f}$ is the shear strength involving fiber failure.

Moreover, a continuous damage evolution model that can characterize the stiffness variation for composites is established. Here, three failure factors, *d*_1_, *d*_2_, and *d*_3_, are used to depict the degree of fiber failure, matrix cracking, and matrix crushing with a domain of [0, 1] as follows:(9)$$d_{1}=1-e^{\left(-E_{11}{\left(\varepsilon_{11}-\varepsilon_{X_{t}}\right)}^{2}\left(\sqrt{e_{f}}-1\right)L^{c}/G_{f}\right)}/\sqrt{e_{f}}$$
(10)$$d_{2}=1-e^{\left(-E_{22}{\left(\varepsilon_{22}-\varepsilon_{Y_{t}}\right)}^{2}\left(\sqrt{e_{mt}}-1\right)L^{c}/G_{m}\right)}/\sqrt{e_{mt}}$$
(11)$$d_{3}=1-e^{\left(-E_{22}{\left(\varepsilon_{22}-\varepsilon_{Y_{c}}\right)}^{2}\left(\sqrt{e_{mc}}-1\right)L^{c}/G_{m}\right)}/\sqrt{e_{mc}}$$
where *L^c^* is the characteristic length, and *G_f_* and *G_m_* denote the fracture energy of the material along the longitudinal and transverse directions, respectively. Therefore, the variation in the stiffness of composites can be depicted as follows:(12)$$\begin{array}{c} C_{11}^{d}=\left(1-d_{1}\right)C_{11},C_{22}^{d}=\left(1-d_{2}\right)C_{22}\\ C_{12}^{d}=C_{21}^{d}=\left(1-d_{1}\right)\left(1-d_{2}\right)C_{12}\\ C_{13}^{d}=C_{31}^{d}=\left(1-d_{1}\right)\left(1-d_{3}\right)C_{13},C_{23}^{d}=C_{32}^{d}=\left(1-d_{2}\right)\left(1-d_{3}\right)C_{23}\\ C_{44}^{d}=\left(1-d_{1}\right)\left(1-d_{2}\right)C_{44},C_{55}^{d}=\left(1-d_{1}\right)\left(1-d_{2}\right)C_{55},C_{66}^{d}=\left(1-d_{1}\right)C_{66} \end{array}$$
where $C_{ij}$ and $C_{ij}^{d}$ represent the stiffness coefficients before and after the onset of the particular damage, respectively.

The initiation of delamination in the applied cohesive elements is determined through the utilization of a mixed-mode secondary stress criterion. The progression of delamination is depicted by employing the coupling secondary critical energy release rate criterion. However, there are some difficulties in determining the interlaminar stiffness since the thickness of the cohesive layer is usually small in the numerical model. In this study, the interlaminar stiffness was obtained by referring to Daudeville et al.’s work [37] and Turon et al.’s work [34], expressed as follows:(13)$$\begin{array}{c} K_{nn}=\frac{\lambda E_{33}}{T}\\ K_{ss}=\frac{2\lambda G_{13}}{T}\\ K_{tt}=\frac{2\lambda G_{23}}{T} \end{array}$$
where *T* is the thickness of the single composite layer, and its value is 0.125 mm in this study. λ denotes an empirical coefficient. After applying the cohesive layers, the equivalent elastic modulus of the laminated plates along the thickness direction can be measured using
(14)$$E_{eff}=\frac{E_{33}}{1+\frac{1}{\lambda }}$$
where λ is taken as 50 to set the difference between $E_{eff}$ and $E_{33}$ as 5%. By referring to De Moura’s work [38], the strengths of the cohesive layers are replaced by in-plane parameters, as follows:(15)$$\begin{array}{l} \sigma_{n}=Y_{t}\\ \sigma_{s}=S_{12}\\ \sigma_{t}=S_{m23} \end{array}$$
where $\sigma_{n}$ is the normal strength of the cohesive layer, and $\sigma_{s}$ and $\sigma_{t}$ are the shear strengths of the cohesive layer. Table 2 presents the orthotropic material properties of the composite lamina and the properties of the cohesive elements governing the interface.

### 4.3. Preloading Step and Impacting Step

In order to continuously simulate the impact process of the composite under preloading conditions, the restart analysis function was employed to realize a preloading step followed by an impacting step in sequence. Firstly, a dynamic explicit step was established to apply in-plane tensile preloads on the laminated plate. In this step, the displacement, defined by the smooth step amplitude curve, was applied to all nodes in the regions that connected with the loading blocks to avoid the influence of an inertia force caused by the discontinuous loading rate [37]. Figure 10 shows the in-plane strain distributions along the horizontal and vertical directions for [0/90]_8_ laminated plates under 1000 με biaxial tensile preloads. The strain distributions of the impact regions are almost smooth and uniform, indicating that the proposed method is feasible and efficient. After checking the value of the biaxial in-plane tensile pre-strains, the following impacting step was conducted based on the calculated result from the previous preloading step by employing restart analysis in the ABAQUS explicit solver.

### 4.4. Validation

To ensure the numerical model’s accuracy, the strain histories of the experiments and numerical results were first compared. As shown in Figure 11, the levels and tendencies of the simulation strain histories were comparatively coincident with the experimental results. Moreover, the delamination areas of the targets between the measurements and computations were also compared to support the further validation of the numerical model, as shown in Figure 12 and Table 3. Since the projection of the delamination area could not be output by ABAQUS, a mini software processed by Python language was designed to calculate the total projected areas of delamination. The work logic of mini Python software is divided into the following steps: output the stiffness degradation field information of each cohesive element layer, compare the scalar stiffness degradation (SDEG) on the same vertical coordinate longitudinally, overwrite the SDEG value circularly, obtain the continuous stiffness degradation field via linear interpolation, and calculate the projected area for stiffness degradation SDEG ≥ 0.999. The numerical results also showed good agreement with the C-scan results, proving that the numerical simulation method applied in this study is appropriate for modeling the delamination behavior of preloaded CFRP laminated plates.

## 5. Discussion

From the experimental and numerical results, it can be found that the preloading value and impact velocity have a significant influence on the delamination behavior of preloaded CFRP laminated plates. In this section, based on the verified numerical simulation method, the effects of the preloading degree and impact velocity on the delamination resistance will be discussed in detail by conducting a series of simulations.

### 5.1. Influence of Biaxial Tensile Pre-Strains Value

Figure 13 shows the numerical delamination areas of CFRP laminated plates under various biaxial tensile preloads when impacting different positions at velocities of 50 m/s, 70 m/s, and 90 m/s. As shown, the delamination areas gradually increase when moving the impact location from the center to the near-edge location. The [0/90]_8_ and [0/90/+45/−45]_2S_ targets under 500 με and 1000 με show more delamination than unloaded targets when impacting the near-edge location at 50 m/s. Figure 14 shows the stiffness degradation of [0/90]_8_ CFRP laminated plates under various biaxial tensile preloads. Using the toggle global translucency function in ABAQUS/Explicit shows that the CFRP laminated plates present different degrees of interface stiffness degradation at different positions when applying various biaxial tensile preloads. The interface stiffness degradations at the near-edge location are significantly higher than those at the center location (almost no interface stiffness degradation), which results in the interfacial strengths of the laminated plates at the near-edge location being weaker than those at the center location. Thus, both the experimental and numerical results indicate that the preloaded targets generate more delamination at the near-edge location in comparison to the center location when suffering impact loading.

It should be noted that the delamination areas of target plates tend to decrease as the biaxial tensile pre-strains increase for all impact locations, from the center location (point 1, off-center distance is 0 mm) to the near-edge location (point 2, off-center distance is 70 mm). In particular, the preloaded targets present relatively fewer delamination areas at the near-edge location than unloaded targets when the preloading value is increased ([0/90]_8_ from 500 με to 1500 με, [0/90/ + 45/−45]_2S_ from 500 με to 1300 με). Since the delamination behavior of preloaded CFRP laminated plates is influenced by the competing mechanisms of the stress-stiffening effect and the interface stiffness degradation effect, it can be considered that the positive effect had on the delamination resistance by stress-stiffening gradually gains an advantage compared to the weakening effect had on the delamination resistance by interface stiffness degradation when the preloading value is increased. With the increase in the biaxial tensile pre-strains value, both the positive effect of stress-stiffening and the weakening effect of interface stiffness degradation are gradually enhanced. However, the enhancement degree of these two competitive factors is different. As can be seen in Figure 6, due to the linear elastic characteristic of composite material, the additional resistance force $2\cdot F_{p}\sin\theta$ induced by the preload is also linear elastic. Thus, the stress-stiffening effect on the delamination resistance is almost linearly enhanced when the biaxial tensile preloads are increased linearly. Figure 15 illustrates the schematic diagram of the stiffness degradation for the interface layer based on the bilinear constitutive model. As shown, the degree of stiffness degradation for the interface gradually reduces when the pre-strain/deformation is increased linearly. Figure 16 also shows the SDEG output by Python software based on the numerical results. As shown, the stiffness degradation degrees at all five assigned locations tend to decrease as the biaxial tensile preloads increase linearly. Therefore, the stress-stiffening effect will gradually gain an advantage over the interface stiffness degradation effect when the biaxial tensile pre-strains value is increased.

### 5.2. Influence of Impact Velocity

Both experimental and numerical results showed that when impacting the near-edge location at impact velocities of 50 m/s, preloaded CFRP laminated plates present more delamination areas than unloaded targets. However, by increasing the impact velocity from 50 m/s to 70 m/s, the preloaded CFRP laminated plates show fewer delamination areas in comparison to unloaded targets, which can also be found in the case of impacting the near-edge location of CFRP laminated plates at the impact velocity of 90 m/s. Figure 17a shows the numerical delamination areas of [0/90]_8_ plates under various biaxial tensile preloads (0 με, 500 με, and 1000 με) when impacting different positions at four different impact velocities (30 m/s, 50 m/s, 70 m/s, and 90 m/s). The preloaded targets present relatively fewer delamination areas at the near-edge location than unloaded targets when the impact velocity is increased to 70 m/s and 90 m/s, which stands in sharp contrast to the case at impact velocities of 30 m/s and 50 m/s. Assuming that the stiffening region of the preloaded CFRP laminated plate is the distribution range and that the delamination areas of the preloaded targets are less than those of unloaded targets, it can be seen that the stiffening regions constantly expand their scopes with the increase in the impact velocity. For example, the stiffening region of the 500 με preloaded target expands its scope from the 30 mm off-center distance to 43 mm off-center distance by increasing the impact velocity from 30 m/s to 50 m/s. In addition, when suffering the 70 m/s impact loading and 90 m/s impact loading, the delamination areas of the preloaded targets are less than those of the unloaded targets at all eight impact locations from the center location to the 70 mm off-center distance. This means that the stiffening regions for both the 500 με and 1000 με preloaded targets expand their scope to all impact locations.

The mechanism of the stress-stiffening effect at various impact velocities is depicted in Figure 17b. It is evident that the out-of-plane displacements of targets subjected to identical biaxial tensile pre-strains increase proportionally with the projectile’s impact velocity, thus resulting in $\theta_{1}<\theta_{2}<\theta_{3}<\theta_{4}$. Therefore, the additional resistance forces induced by biaxial tensile preloads also increase as the impact velocity increases, depicted as follows:(16)$$2\cdot F_{t}\sin\theta_{1}<2\cdot F_{t}\sin\theta_{2}<2\cdot F_{t}\sin\theta_{3}<2\cdot F_{t}\sin\theta_{4}$$

Therefore, as the impact velocity increases, the positive effect of stress stiffening on the delamination resistance gradually enhances while the weakening effect of interface stiffness degradation on delamination resistance remains the same, indicating that the stress-stiffening effect gains an advantage over the interface stiffness degradation effect when competing to have an effect on delamination resistance.

## 6. Conclusions

An extensive investigation was conducted by combining ballistic impact tests, dynamic finite element simulations, and theoretical analysis to further understand the impact delamination behaviors of CFRP laminated plates under biaxial tensile preloads. The main conclusions were as follows:(1)Both the experimental findings and simulations consistently illustrated that biaxial tensile preloading could bolster resistance to out-of-plane displacement, thereby exerting a beneficial influence on the delamination resistance of CFRP laminated plates (14.2~36.7% decrease in delamination areas under 1000 με). However, no more than a 19.3% decrease in delamination areas was observed when impacting the near-edge location, and the case under impact velocity of 50 m/s even showed increasing delamination areas. This phenomenon indicates that applying preloads was also supposed to induce the interface stiffness degradation effect, which was considered a negative effect on delamination resistance.(2)The impact velocity can influence the competing mechanisms of the stress-stiffening effect and the interface stiffness degradation effect. Since the degree of interface stiffness degradation for CFRP laminated plates with specific stacking sequences is the same when applying the same biaxial tensile preloads, the influence of impact velocity on the competing mechanisms of the stress-stiffening effect and the interface stiffness degradation effect is mainly caused by the influence of impact velocity on the stress-stiffening effect.(3)In near-edge location ballistic tests, it was noted that the preloaded CFRP laminated plates exhibited increased delamination compared to the unloaded target at an impact velocity of 50 m/s. However, at impact velocities of 70 m/s and 90 m/s, the preloaded CFRP laminate plates displayed reduced instances of delamination in comparison to the unloaded target. It can be concluded that the stress-stiffening effect becomes progressively more dominant than the interface stiffness degradation effect as the preloading value and impact velocity increase.(4)A conceptual framework involving the competing mechanisms of the stress-stiffening effect and the interface stiffness degradation effect has been formulated to elucidate the impact of biaxial in-plane tensile preloads on delamination behavior, and such a competitive mechanism was found to be influenced by the preloading value and impact velocity. With an increase in the biaxial tensile pre-strains value or impact velocity, the stress-stiffening effect gradually gained an advantage over the interface stiffness degradation effect. Considering the effect of the preloading degree and impact velocity on the competitive mechanism, the complex delamination behaviors at different impact locations of CFRP laminated plates under various preloading degrees and impact velocities were reasonably explained.

## Figures and Tables

**Figure 1 materials-16-06595-f001:**
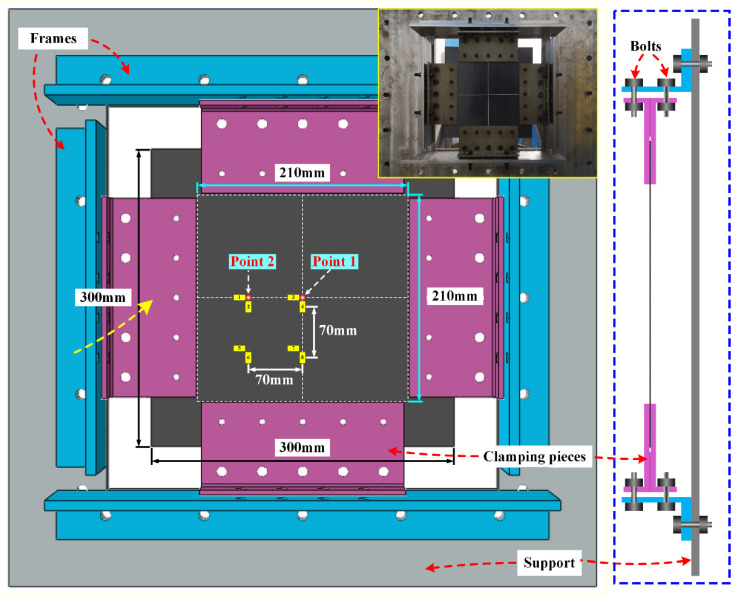
Configuration of preloading fixture and strain measurement scheme.

**Figure 2 materials-16-06595-f002:**
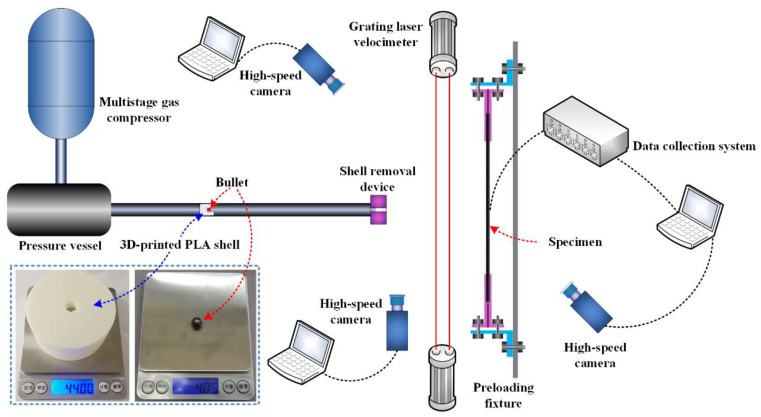
Schematic representation of a single-stage compressed gas gun system.

**Figure 3 materials-16-06595-f003:**
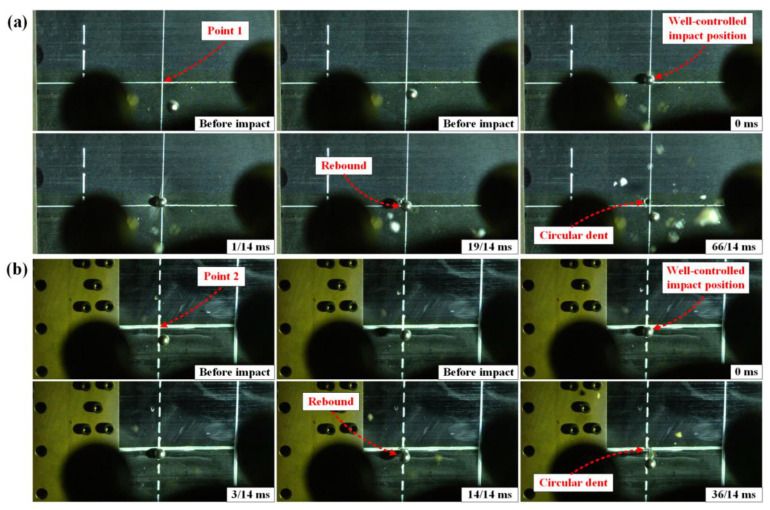
Typical high-speed photographs of CFRP laminated plates impacted by sphere steel projectile under the speed of 50 m/s; (**a**) impact on point 1 (center location); (**b**) impact on point 2 (near-edge location).

**Figure 4 materials-16-06595-f004:**
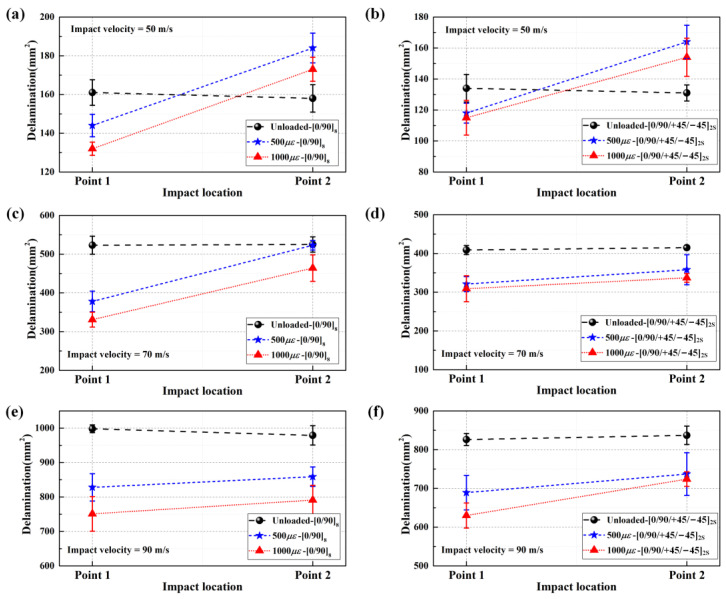
Delamination areas of CFRP laminated plates under various biaxial tensile preloads at impact speeds of 50 m/s, 70 m/s, and 90 m/s; (**a**,**b**) delamination areas of [0/90]_8_ plates and [0/90/+45/−45]_2S_ plates under 50 m/s; (**c**,**d**) delamination areas of [0/90]_8_ plates and [0/90/+45/−45]_2S_ plates under 70 m/s; (**e**,**f**) delamination areas of [0/90]_8_ plates and [0/90/+45/−45]_2S_ plates under 90 m/s.

**Figure 5 materials-16-06595-f005:**
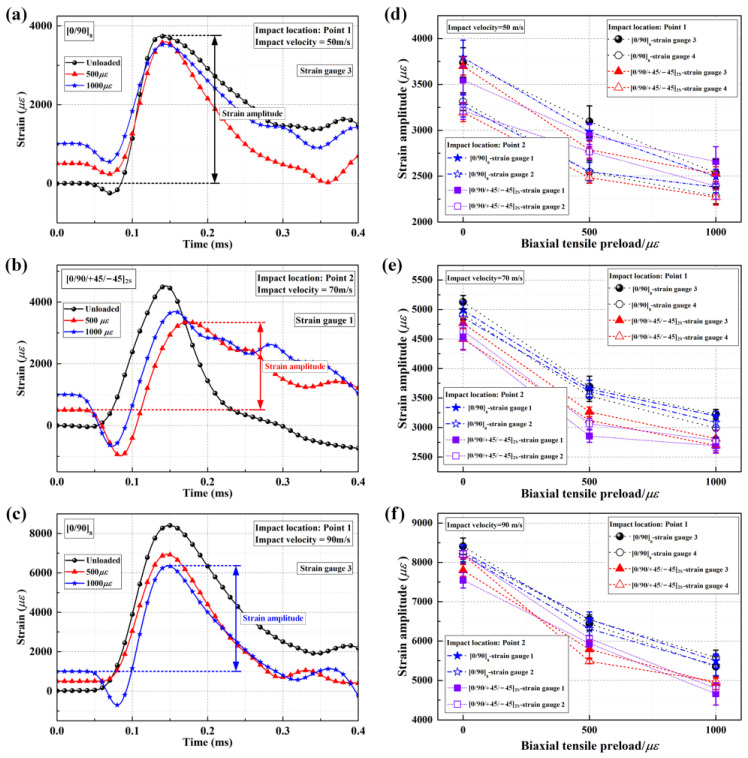
In-plane strain histories and strain amplitudes of CFRP laminated plates under various biaxial tensile preloads; (**a**–**c**) typical in-plane strain histories at impact speeds of 50 m/s, 70 m/s, and 90 m/s; (**d**–**f**) strain amplitudes calculated from in-plane strain histories at impact speeds of 50 m/s, 70 m/s, and 90 m/s.

**Figure 6 materials-16-06595-f006:**
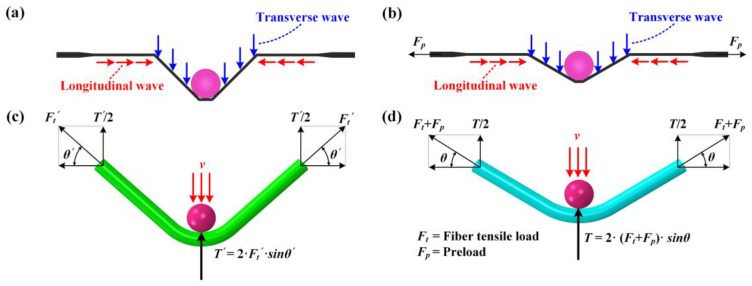
Enhancement mechanism of resistance to out-of-plane displacement for CFRP laminated plates when applying in-plane tensile preloads; (**a**,**b**) “V” shape caused by longitudinal wave and transverse wave with and without tensile preloads; (**c**,**d**) bending degree of plates with and without tensile preloads.

**Figure 7 materials-16-06595-f007:**
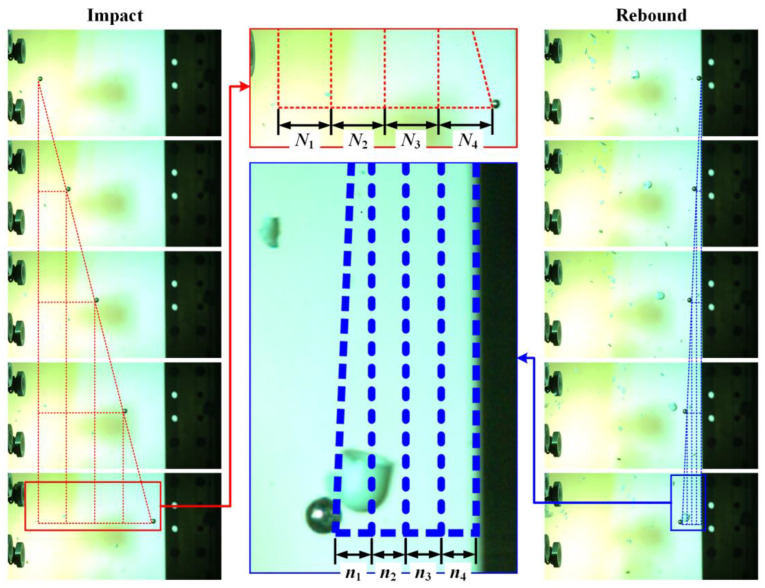
Typical photographs, taken with the high-speed camera, of the impact process and rebound process of the projectile.

**Figure 8 materials-16-06595-f008:**
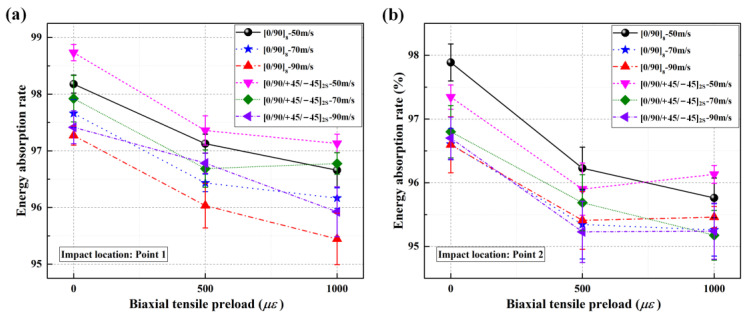
Energy absorption rates of CFRP laminated plates under various biaxial tensile preloads at impact speeds of 50 m/s, 70 m/s, and 90 m/s; (**a**) energy absorption rates when impacting point 1; (**b**) energy absorption rates when impacting point 2.

**Figure 9 materials-16-06595-f009:**
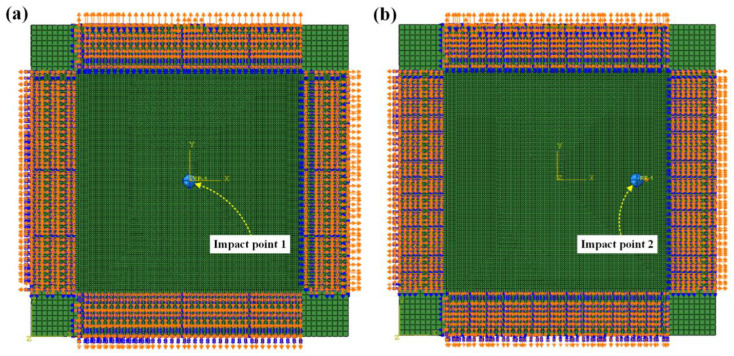
Mesh for L × W = (300 × 300) mm^2^ laminated plate; (**a**) impact on point 1 (center location); (**b**) impact on point 2 (near-edge location).

**Figure 10 materials-16-06595-f010:**
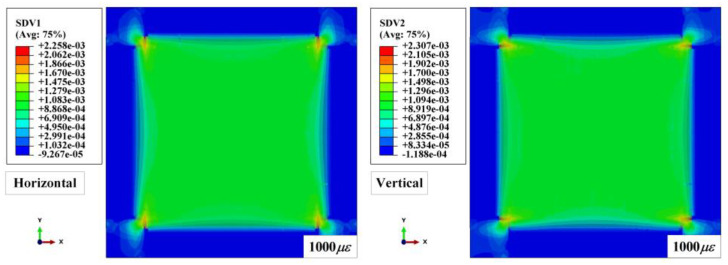
In-plane strain distributions along the horizontal and vertical direction of preloaded [0/90]_8_ laminated plates.

**Figure 11 materials-16-06595-f011:**
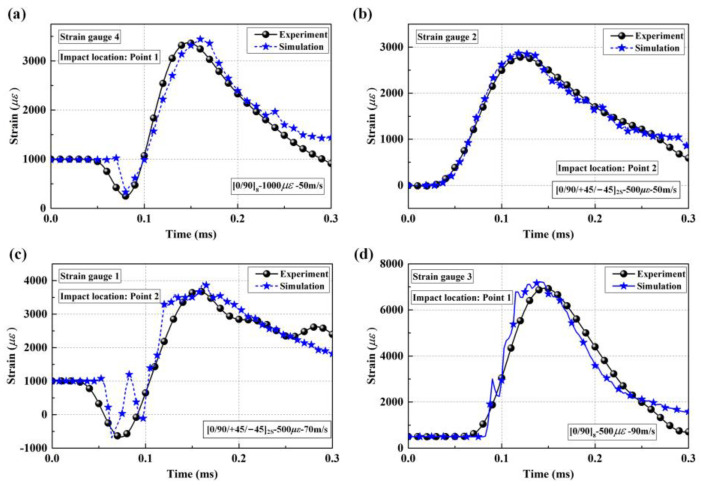
Comparison of typical strain histories between experiments and simulations; (**a**) strain gauge 4 on [0/90]_8_ plate; (**b**) strain gauge 2 on [0/90/+45/−45]_2S_ plate; (**c**) strain gauge 1 on [0/90/+45/−45]_2S_ plate; (**d**) strain gauge 3 on [0/90]_8_ plate.

**Figure 12 materials-16-06595-f012:**
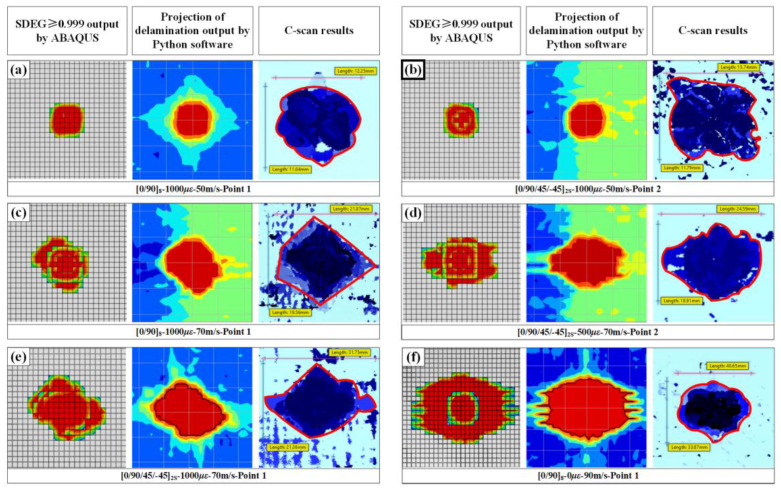
Comparison of typical delamination between experiments and simulations; (**a**) impact on point 1 of 1000 με preloaded [0/90]_8_ plates at speeds of 50 m/s; (**b**) impact on point 2 of 1000 με preloaded [0/90/+45/−45]_2S_ plates at speeds of 50 m/s; (**c**) impact on point 1 of 1000 με preloaded [0/90]_8_ plates at speeds of 70 m/s; (**d**) impact on point 2 of 500 με preloaded [0/90/+45/−45]_2S_ plates at speeds of 70 m/s; (**e**) impact on point 1 of 1000 με preloaded [0/90/+45/−45]_2S_ plates at speeds of 70 m/s; (**f**) impact on point 1 of unloaded [0/90]_8_ plates at speeds of 90 m/s.

**Figure 13 materials-16-06595-f013:**
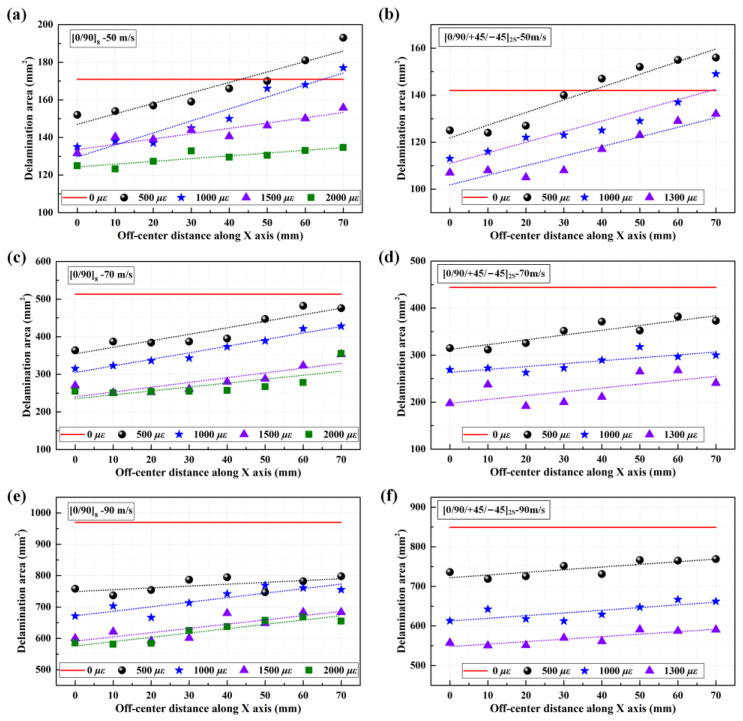
Numerical delamination areas of CFRP laminated plates under various biaxial tensile preloads when impacting different positions. (**a**,**b**) delamination areas of [0/90]_8_ plates and [0/90/+45/−45]_2S_ plates under 50 m/s; (**c**,**d**) delamination areas of [0/90]_8_ plates and [0/90/+45/−45]_2S_ plates under 70 m/s; (**e**,**f**) delamination areas of [0/90]_8_ plates and [0/90/+45/−45]_2S_ plates under 90 m/s.

**Figure 14 materials-16-06595-f014:**
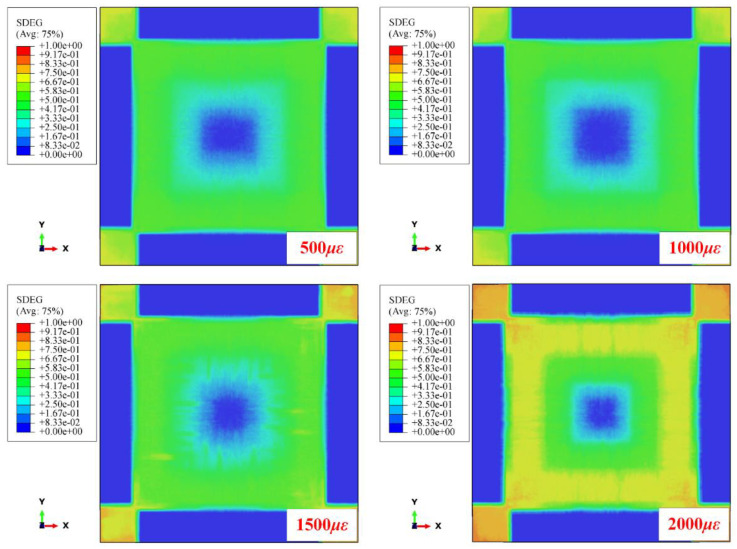
Stiffness degradation of [0/90]_8_ CFRP laminated plates under various biaxial tensile preloads.

**Figure 15 materials-16-06595-f015:**
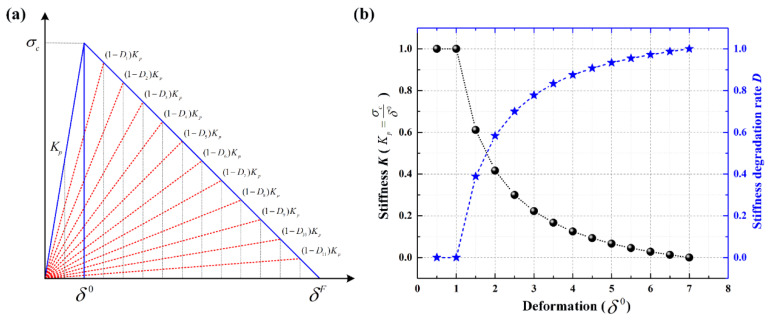
Schematic diagram of stiffness degradation for interface layer based on bilinear constitutive model; (**a**) bilinear constitutive model; (**b**) stiffness degradation for the interface layer.

**Figure 16 materials-16-06595-f016:**
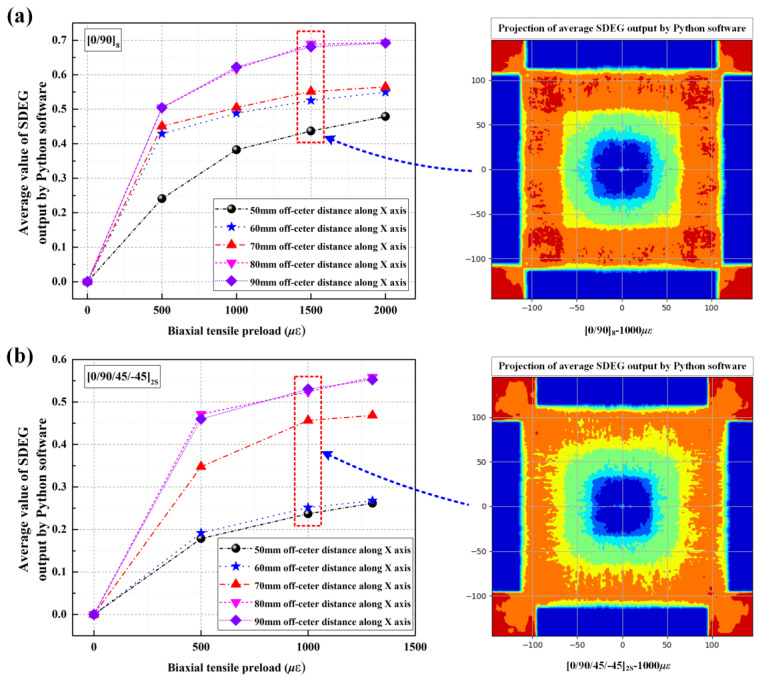
Stiffness degradations output by Python software based on simulations; (**a**) [0/90]_8_ plates; (**b**) [0/90/+45/−45]_2S_ plates.

**Figure 17 materials-16-06595-f017:**
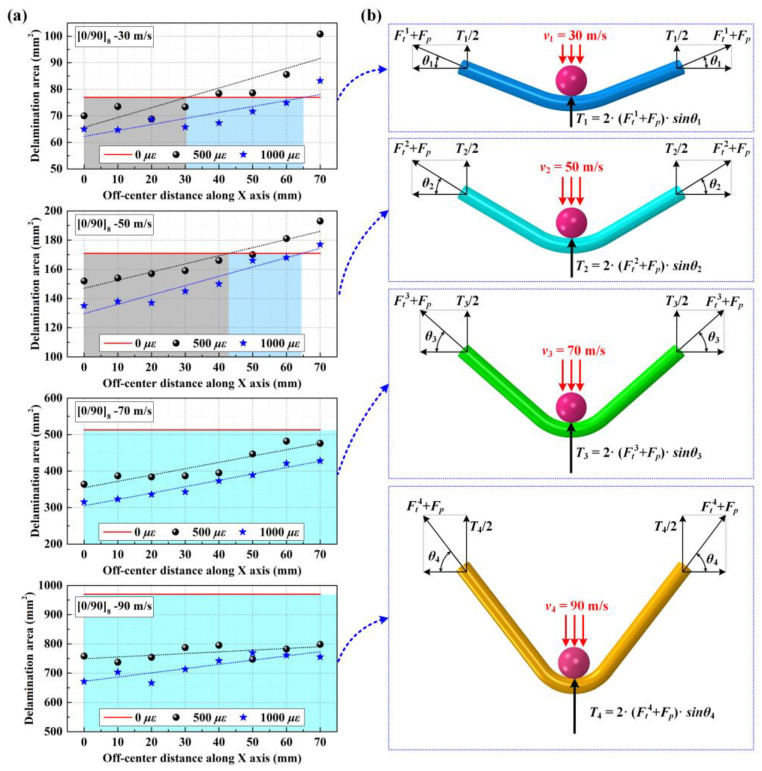
Numerical delamination areas and stress-stiffening effect mechanism under different impact velocities; (**a**) numerical delamination areas when impacting different positions of [0/90]_8_ plates under different impact velocities; (**b**) stress-stiffening effect mechanism under different impact velocities.

**Table 1 materials-16-06595-t001:** Delamination areas from C-scan experiments.

Layups	Impact Location	Pre-Strain	Delamination Areas (mm^2^)		
			*V* = 50 m/s	*V* = 70 m/s	*V* = 90 m/s
[0/90]_8_	Point 1	0 με	161 ± 6.6	523 ± 23.2	998 ± 11.2
		500 με	144 ± 5.8	378 ± 26.5	828 ± 39.7
		1000 με	132 ± 3.4	331 ± 19.3	751 ± 50.4
	Point 2	0 με	158 ± 7.1	525 ± 19.7	979 ± 28.1
		500 με	184 ± 7.7	523 ± 12.3	859 ± 28.3
		1000 με	173 ± 6.2	464 ± 34.1	791 ± 42.9
[0/90/+45/−45]_2S_	Point 1	0 με	134 ± 8.9	409 ± 11.8	826 ± 15.5
		500 με	118 ± 6.4	321 ± 19.1	689 ± 44.6
		1000 με	115 ± 11.2	309 ± 33.6	630 ± 32.2
	Point 2	0 με	131 ± 5.2	415 ± 6.2	837 ± 23.7
		500 με	164 ± 10.7	358 ± 38.7	737 ± 55.2
		1000 με	154 ± 12.3	337 ± 11.4	724 ± 18.8

**Table 2 materials-16-06595-t002:** Orthotropic material properties for composite lamina and cohesive element properties for interface.

Materials	Parameters	Values
Composite lamina	Density	*ρ* = 1510 kg/m^3^
	Young’s modulus	*E*_11_ = 151.8 GPa, *E*_22_ = 12 GPa
		*G*_12_ = *G*_13_ = 3.3 GPa, *G*_23_ = 2.0 GPa
	Poisson’s ratio	*ν*_12_ = *ν*_13_ = 0.03, *ν*_23_ = 0.38
	Strength	*X_t_* = 1872 MPa, *Y_c_* = 150 MPa, *Y_t_* = 34 MPa
		*S*_12_ = *S_m_*_23_ = 100 MPa, *S_f_* = 160 MPa
	Fracture energy	*G_f_* = 92,000 J/m^2^, *G_m_* = 600 J/m^2^
Interface	Density	*ρ* = 1000 kg/m^3^
	Stiffness	*K_nn_* = 4.8 × 10^6^ N/mm^3^, *K_ss_* = *K_tt_* = 2.64 × 10^6^ N/mm^3^
	Strength	*σ_n_* = 34 MPa, *σ_s_* = *σ_t_* = 100 MPa
	Fracture energy	*G_IC_* = 600 J/m^2^, *G_IIC_* = *G_IIIC_* =1200 J/m^2^

**Table 3 materials-16-06595-t003:** Comparison of delamination areas between experiments and simulations.

Layups Impact Location	Pre-Strain	Numerical Delamination Area (mm^2^) and Errors					
		V = 50 m/s	Errors	V = 70 m/s	Errors	V = 90 m/s	Errors
[0/90]_8_ Point1/Point 2	0 με	171	6.2%	513	−1.9%	970	−2.8%
	500 με	152	5.5%	364	−3.7%	758	−8.5%
	1000 με	135	2.3%	315	−5.1%	671	−10.7%
	0 με	171	8.2%	513	−2.3%	970	−0.9%
	500 με	193	4.9%	476	−9.0%	798	−7.1%
	1000 με	177	2.3%	428	−7.8%	755	−4.6%
[0/90/+45/−45]_2S_ Point1/Point 2	0 με	142	6.0%	444	8.6%	849	2.8%
	500 με	125	5.9%	315	−1.9%	736	6.8%
	1000 με	113	−4.9%	269	−12.9%	613	−2.7%
	0 με	142	7.6%	444	7.0%	849	1.4%
	500 με	156	−4.9%	373	4.2%	769	4.3%
	1000 με	149	2.3%	300	−11.0%	662	−8.6%

## Data Availability

The data are available from the corresponding author on reasonable request.

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
