# Peer review of "Delamination Behavior of CFRP Laminated Plates under the Combination of Tensile Preloading and Impact Loading"

_materials, 2023, doi:10.3390/ma16196595_

Round 1
Reviewer 1 Report
All the reviewer's Comments are addressed on the file attached.

Moderate editing of the English language required
Reviewer 2 Report
- In the abstract,
Have to include mentioning the important comparative results of this paper related to delamination and the known values (by numbers), and insert the percentage of convergence or similarity between them (experiments and simulations), this enhances the correct scientificity of the paper.
- Have to write the values of the variables between parentheses and mention their units outside the parentheses.
For example (300x300) mm2 … etc.
Lines (82, 206, 257 and 315)
Rewrite the sentences (our previous work) Without using pronouns
Line 14:
Rewrite the sentences (to enhance our understanding, we conducted numerical simulations using......etc.) without using pronouns
In lines (168 and 169) move it to conclusion
In lines (239, 240, 241, 242 and 2430 move to conclusion
In lines (273, 274, 275, 276, 277, 278,279 and 280) rewrite (In brief) and move it to conclusion
In lines (481, 482, 483, 484, 485, 486 and 487) rewrite (In brief) and move it to conclusion
In lines (168, 239, 273 and 481)
Have to remove the sentences (it can be concluded) or rewrite the editing of the sentence.
In line 503
-Rewrite the conclusion
- Have to remove words ( summary and) and stay just (Conclusions)
Reviewer 3 Report
The main question manuscript addresses delamination behavior of CFRP laminated plates under the combination of tensile preloading and impact loading.
1. The methodology adopted for study is correct and needs no further improvement and controls. However, I suggest to provide detailed explanation for readers of the journal.
2. The conclusion section needs to be improved to align with evidences presented in the manuscript as well as the main question addressed in the manuscript.
3. References need to be improved by adding the pertinent literature in the introduction section.
4. Abstract should be improved, including the novelty of this work, Result and discussion section should be provided with more graphical representation or figures with technical interpretation.
Reviewer 4 Report
Authors need to address the following points to improve the quality of the article.
1. Bibliography numbers do not match with citation numbers. Kindly rectify the same.
2. The abstract needs to be improved significantly with quantitative data.
3. Line 44 – “Yew and Kendrick et al. [4]” When two authors are there, then et al., should not be used. Check the same in the entire manuscript.
4. Emphasize how the study filled a knowledge gap or addressed a specific issue.
5. In Table 1: Mention Unit of “Delamination areas”.
6. Kindly reconcile the conclusion with the study objectives.
7. What are the practical implications of this study and the future directions? Kindly state?
8. The article must undergo language proofreading, as I found grammatical errors.
The article must undergo language proofreading, as I found grammatical errors.
